# ImDy: Human Inverse Dynamics from Imitated Observations

**Xinpeng Liu**[1,2]**, Junxuan Liang**[1]**, Zili Lin**[1]**, Haowen Hou**[3]**, Yong-Lu Li**[1,2]***Cewu Lu**[1,2]*
[1]Shanghai Jiao Tong University, [2]Shanghai Innovation Institute, [3]Soochow University
xinpengliu0907@gmail.com, {whitefork,linzili111666}@sjtu.edu.cn
haowenhou@outlook.com, {yonglu_li,lucewu}@sjtu.edu.cn

## ABSTRACT

Inverse dynamics (ID), which aims at reproducing the driven torques from human kinematic observations, has been a critical tool for human motion analysis. However, it is hindered from wider application to general motion due to its limited scalability. Conventional optimization-based ID requires expensive laboratory setups, restricting its availability. To alleviate this problem, we propose to exploit the recently progressive human motion imitation algorithms to learn human inverse dynamics in a *data-driven* manner. The key insight is that the human ID knowledge is implicitly possessed by motion imitators, though not directly applicable. In light of this, we devise an efficient data collection pipeline with state-of-the-art motion imitation algorithms and physics simulators, resulting in a large-scale human inverse dynamics benchmark as **Imitated Dynamics** (**ImDy**). ImDy contains over **150 hours** of motion with joint torque and full-body ground reaction force data. With ImDy, we train a data-driven human inverse dynamics solver **ImDyS(olver)** in a fully supervised manner, which conducts ID and ground reaction force estimation simultaneously. Experiments on ImDy and real-world data demonstrate the impressive competency of ImDyS in human inverse dynamics and ground reaction force estimation. Moreover, the potential of ImDy(-S) as a fundamental motion analysis tool is exhibited with downstream applications. The project page is https://foruck.github.io/ImDy.

## 1 INTRODUCTION

The rapid progress in human motion capture based on computer vision has made an enormous amount of human motion data available to the research community (Mahmood et al., 2019; Mandery et al., 2016). The accumulation of human motion manages to push motion understanding forward in various tasks, including behavior understanding (Punnakkal et al., 2021; Shahroudy et al., 2016) and character animation (Guo et al., 2022; Tevet et al., 2023; Liu et al., 2025a;b). However, given the vision-based nature, most current efforts focus only on visible kinematics information. The invisible factors, especially the dynamic factors, which could carry deeper insights into the underlying production mechanism of human motion, are typically overlooked, such as *driven torques* and *ground reaction forces*. This limits the current motion understanding algorithms from wider applications to domains where physical constraints must be seriously considered, such as robotics (Figueredo et al., 2020; Teramae et al., 2017), healthcare (Yao et al., 2018), and sports training (Caruntu & Moreno, 2019). To alleviate this, we focus on identifying the driven torques and ground reaction forces for human motion from pure kinematics MoCap data, known as human inverse dynamics (ID).

Human inverse dynamics, as a basic step toward physical motion modeling, has been extensively discussed by the biomechanics community for applications like gait analysis. A fundamental obstacle is that it could not be measured non-intrusively. Therefore, computationally expensive optimization-based methods are widely adopted and mature software is developed (Delp et al., 2007; Damsgaard et al., 2006; Werling et al., 2021). However, accurately measured ground reaction forces are required to ensure a determinate solution, which could be expensive and applicable only in restricted

---
*Corresponding authors.

laboratory settings. Also, the optimization process could be sensitive to small disturbances in either motion capture noises or subject variances. These make it hard to scale up for wider applications to general motion. Given the success achieved by data-driven methods in CV and NLP, deep-learning-based methods are proposed (Zell & Rosenhahn, 2015; Zell et al., 2017; Lv et al., 2016), aiming at scalable human inverse dynamics with only kinematic observations as inputs.

Unfortunately, *data acquisition* becomes a major bottleneck since laboratory setups are still required for ground-truth acquisition.

Given this, we project our sights on the recent progress of Imitation Learning (IL) (Luo et al., 2021; 2023), which replicates recorded human motion through fully simulated humanoids with physical control signals, namely, joint torques. A key insight is that with the goal of kinematics phenomenon imitation, IL might also *implicitly* imitate the dynamics production mechanism, known as ID. However, IL is not directly applicable to ID. Despite the visual resemblances between the recorded and simulated motion, kinematic errors still exist. These errors could be neglected for kinematic analyses, however, for dynamic analysis, they could be amplified drastically (Uchida & Seth, 2022). Moreover, existing successful IL algorithms are typically based on joint-actuated SMPL (Loper et al., 2015) avatars, whose physical properties and topol-

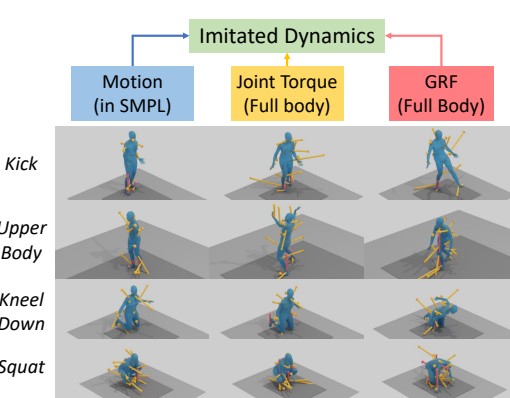

Figure 1: ImDy pairs diverse SMPL motion data with dynamics including full-body torques and ground reaction forces (GRF) like the right knee GRF for kneeling, which could be hard to achieve under conventional laboratory setups.

ogy differ from real humans. To this end, extracting ID knowledge from IL becomes critical. Here, we adopt the state-of-the-art motion IL algorithm (Luo et al., 2023) and physics simulator (Makoviychuk et al., 2021) to imitate recorded motions, extracting the observed kinematic states, joint torques, and the ground reaction forces, resulting in a large-scale human inverse dynamics database named **Im**itated **Dy**namics (ImDy) with more than 150-hour human motion. There are two major merits of ImDy. First, it is *scalable*. Multiple samples could be concurrently collected in the simulator without expensive laboratory setups, extending the border of ID data acquisition. As shown in Fig. 1, we could even pair some rather complex motions with ID data, which is hard to achieve in laboratories. Second, it is *holistic*. Beyond the ground reaction force and ID typically recorded in laboratories for previous efforts (Zell et al., 2020; Mourot et al., 2022; Han et al., 2023), the physics simulator enables us to access the GRFs and joint torques of all human body segments, as shown in Fig. 1.

With the accumulated data, we could address the human inverse dynamics in a fully supervised manner. Given the observed kinematics states that describe a motion transition in a certain period, we train a data-driven solver as ImDyS(olver) to estimate the ground reaction forces and the internal dynamics to drive the transition. We also devise losses to regulate ImDyS with forward dynamics awareness and motion plausibility constraints.

We demonstrate the efficacy of ImDyS through a wide span of experiments. First, we evaluate our method on ImDy for a basic performance illustration with simulated ImDy. Then ImDyS is evaluated on GroundLink (Han et al., 2023), which contains real-world ground reaction force. Furthermore, we demonstrate the efficacy of ImDy on the recent real-world human dynamics dataset AddBiomechanics (Werling et al., 2025).

Our contribution could be summarized as: (1) We propose a novel pipeline for human inverse dynamics data collection, introducing a large-scale benchmark as ImDy. (2) Based on ImDy, a data-driven ID solver is instantiated as ImDyS. (3) Extensive experiments are conducted with analyses of the proposed data-driven methodology, demonstrating the feasibility of ImDyS.

## 2 BACKGROUND

**Conventional Inverse Dynamics.** Inverse dynamics, known as inferring forces/moments from kinematic observations, have been discussed for long in the biomechanics community. In this literature,

it is formulated as an optimization problem: given a representative model of a subject, the joint kinematics over time w.r.t. the subject model, and the external forces, find the driving torques that produce the motion (Uchida & Delp, 2021). The Newtonian dynamic equations are involved as

$$M(q)\ddot{q} + C(q, \dot{q}) + G(q) = J\lambda + \tau, \tag{1}$$

where $M(q)$ is the generalized human inertia matrix w.r.t. generalized coordinate $q$, $C(q, \dot{q})$ is the Coriolis and centrifugal forces, $G(q)$ represents gravity, $J$ is the Jacobian matrix mapping external forces $\lambda$ to the generalized coordinates. Thus, the driven torques $\tau$ could be obtained by minimizing the difference between the left and right terms of Eq. 1. Mature software based on this has been developed like OpenSim (Delp et al., 2007), AnyBody (Damsgaard et al., 2006), and Nimble (Werling et al., 2021). In addition, many efforts are made for clinical motion analysis (Fukuchi et al., 2018; Schreiber & Moissenet, 2019). However, these efforts are not as extensively recognized by the computer vision and computer graphics community as expected due to the scalability issue. Despite the elegant formulation, the efficacy of optimization-based heavily relies on the quality of external force $\lambda$ (like GRF) measurement, whose cost could be non-trivial. Therefore, most of them focused on limited motion in laboratory settings. Some resort to wearable devices (Latella et al., 2016; 2019) to partially mitigate the limitation. In addition, fitting the raw captured kinematic observations to a specific human model for joint kinematics could be time-consuming and unstable, even with recent progress on it (Keller et al., 2023; Werling et al., 2023).

**Learning-based Inverse Dynamics.** With the progress in deep learning, there have been efforts to adopt neural networks to address the human ID problem. Many efforts focus on lower-body-only (Johnson & Ballard, 2014; Xiong et al., 2019) or upper-body-only (Manukian et al., 2023) inverse dynamics. More recently, Lv et al. (2016) collected over 1 hour of motion with an optical MoCap system, four force plates, and a pair of pressure insoles. The ground truth was obtained through optimization and a Gaussian mixture framework was devised. Zell & Rosenhahn (2015); Zell et al. (2017); Zell & Rosenhahn (2017) introduced a predictive dynamics-based human modeling for the acquisition of ground truth. Hundreds of motions were collected and different data-driven techniques were adopted for joint torque regression. Zell et al. (2020) proposed a weakly supervised method based only on motion for gait analysis. These efforts were constrained by costly data acquisition in real-world scenarios, resulting in limited data scale. Very recently, Werling et al. (2025) aggregated multiple existing biomechanics datasets, considerably boosting the data scale. However, most of the collected sequences contained only regular exercise motion with limited diversity. Some efforts focused on ground reaction forces such as (Rempe et al., 2020; Scott et al., 2020), UnderPressure (Mourot et al., 2022), and GroundLink (Han et al., 2023). Some recent works incorporated inverse dynamics into vision-based markerless MoCap systems. Shimada et al. (2021) and Li et al. (2022) simultaneously captured motion and joint torques with customized fully differentiated pipelines. A series of works (Yi et al., 2022; Gartner et al., 2022; Gärtner et al., 2022; Huang et al., 2022; Wang et al., 2023) imitated the captured motion in physical simulators with PD controllers and obtained the torques. However, an inherent problem is the amplification effect from kinematic errors to dynamic errors. As measured by Uchida & Seth (2022), only a 2-cm uncertainty of marker placement in a marker-based MoCap system could result in a peak ankle plantarflexion moment of 26.6 $N \cdot m$. Considering the precision of current markerless MoCap algorithms, the accuracy of the accompanied inverse dynamics could be questionable. Also, among all these efforts for learning-based inverse dynamics, only a few (Zell et al., 2017; Zell & Rosenhahn, 2017; Zell et al., 2020) were quantitatively evaluated with limited locomotion data. A scalable benchmark for learning-based inverse dynamics is still not available.

**Motion Imitation.** IL for human motion replicates recorded human motion sequences with physically controlled simulated characters, which could be inherently close to ID. Most early efforts focus on specified usages with limited generalizability (Bergamin et al., 2019; Peng et al., 2021; Won et al., 2021; 2022; Peng et al., 2022). With residual force control (Yuan & Kitani, 2019), which imposed supernatural forces at the root joint of the humanoid, Luo et al. (2021) generalized to 97% sequences in AMASS (Mahmood et al., 2019). Luo et al. (2023) eliminated the supernatural root force and achieved a 98.9% success rate on AMASS with fall-state recovery. The progress in human motion IL makes it possible to collect human-like motions with full dynamics, shedding new light on the scalable human ID data collection.

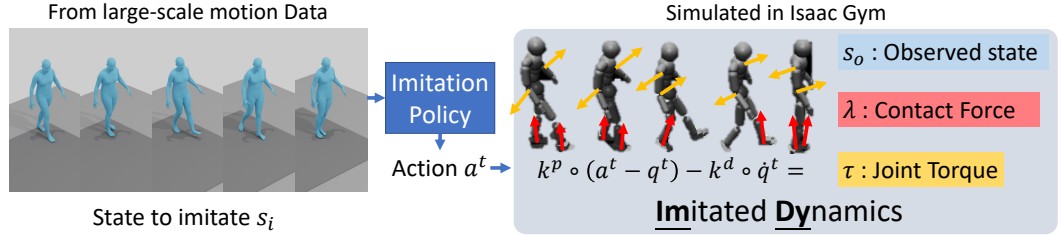

Figure 2: ImDy construction. We first train a motion imitation policy following Luo et al. (2023). Then, the policy is adopted to imitate arbitrary motions, with the imitated states recorded as ImDy.

## 3 CONSTRUCTING IMITATED DYNAMICS

ImDy aims to exploit the inherent closeness of inverse dynamics and imitation learning. Generally, the inverse dynamics (ID) and imitation algorithms (IL) could be abstracted as

$$\tau = ID(s_o^t, s_o^{t+1}), \ \tau = IL(s_o^t, s_i^{t+1}), \tag{2}$$

with driven torque $\tau$, timestamp $t$, observed kinematic states $s_o$, and the state to imitate $s_i$. Both ID and IL learn the dynamic production mechanism of human motion. However, IL algorithms are not directly applicable to ID due to the non-equivalence between $s_o^{t+1}$ and $s_i^{t+1}$. The errors in kinematics could be magnified in dynamics (Uchida & Seth, 2022). This also makes ID algorithms that are deeply coupled with markerless MoCap less reliable. However, it is possible to extract knowledge from IL for ID. In this section, we introduce a simple but effective ID data collection pipeline with IL algorithms. First, the adopted IL algorithm (Luo et al., 2023) is briefly covered in Sec. 3.1. Then, the data collection pipeline is introduced in Sec. 3.2. An overview is given in Fig. 2.

### 3.1 IMITATION LEARNING BASICS

A motion imitator $\pi(a^t | s_o^t, s_i^t)$ is trained following Luo et al. (2023) to solve the Markov Decision Process $\mathcal{M} = \langle \mathcal{T}, \mathcal{S}, \mathcal{A}, \mathcal{R}, \gamma \rangle$. The transition dynamics $\mathcal{T}$ and states $\mathcal{S}$ are governed by the physics simulator. For each timestamp $t$, the policy $\pi$ produces action $a^t \in \mathcal{A}$ and the reward $\mathcal{R}$, based on state $s \in \mathcal{S}$. The training goal is maximizing the reward expectation $\mathbf{E}(\sum_{t=1}^{T} \gamma^{t-1} r^t)$.

**Transition.** IsaacGym (Makoviychuk et al., 2021) is adopted for simulation. A 24-joint humanoid with SMPL (Loper et al., 2015) kinematics and physical properties following Luo et al. (2021; 2023) is adopted with variable shape parameter $\beta \in \mathbb{R}^{10}$. Thus, a human pose at timestamp $t$ could be defined as $q^t = \{\theta^t, p^t\}$, where $\theta^t \in \mathbb{R}^{J \times 6}$ is the joint rotation in the 6d representation (Zhou et al., 2019) and $p^t \in \mathbb{R}^{J \times 3}$ is the 3D joint position.

**State.** At timestamp $t$, $s^t$ contains the observed $s_o^t$ and $s_i^{t+1}$ to imitate. $s_o^t$ is defined in simulator as $s_o^t = (q_t, \dot{q}_t, \beta)$ with 3D body pose $q_t$, velocity $\dot{q}_t$, and body shape $\beta$. $s_i^{t+1}$ is defined similarly except that it is the reference motion with finite-differentiated velocities.

**Action.** All joints but the pelvis are actuated with proportional derivative (PD) controllers, with $a^t$ as the PD target. The torque applied could be calculated as

$$\tau^t = k^p \circ (a^t - q^t) - k^d \circ \dot{q}^t. \tag{3}$$

**Reward.** The reward is composed of four terms: motion imitation reward for minimizing the difference between the imitated states and the expected states, fail-state recovery reward (Luo et al., 2023), AMP reward (Peng et al., 2021), and energy reward to reduce jittering.

**Training.** Following PHC, three primitive policies are progressively trained with hard negative mining, two for pure motion imitation, and one for fail-state recovery. Then, a composer learns to combine the primitives dynamically. PPO (Schulman et al., 2017) is adopted to train the policies.

### 3.2 IMITATED DATA ACQUISITION

With the imitator $\pi$, we pursue to extract its inherent ID knowledge. As in Eq. 2, though the imitator-produced $\tau$ is not accurate for $s_o^t \rightarrow s_i^{t+1}$ since $s_i^{t+1}$ is not guaranteed to reach, $\tau$ is accurate for

Table 1: ImDy compared to related human dynamics datasets. Zell et al. (2020) recorded full-body data but simplified the upper body with a single torso segment. All previous efforts contain only GRF for feet (indicated with *), while we include full body GRF.

| Dataset | #Subj | Duration (h) | Dynamics | Body Repr. | Style |
|---|---|---|---|---|---|
| Zell et al. (2020) | 22 | 0.07 | GRF* & Torques | Partial Skeleton* | Real |
| Scott et al. (2020) | 10 | 7.6 | vertical GRF* | Skeleton | Real |
| UnderPressure (Mourot et al., 2022) | 10 | 5.5 | vertical GRF* | Skeleton | Real |
| GroundLink (Han et al., 2023) | 7 | 1.5 | GRF* | SMPL | Real |
| AddBiomechanics (Werling et al., 2025) | 273 | 57.6 | GRF* & Torques | Rajagopal et al. (2016) | Real |
| ImDy | **435** | **152.3** | **GRF & Torques** | **SMPL** | Simulated |

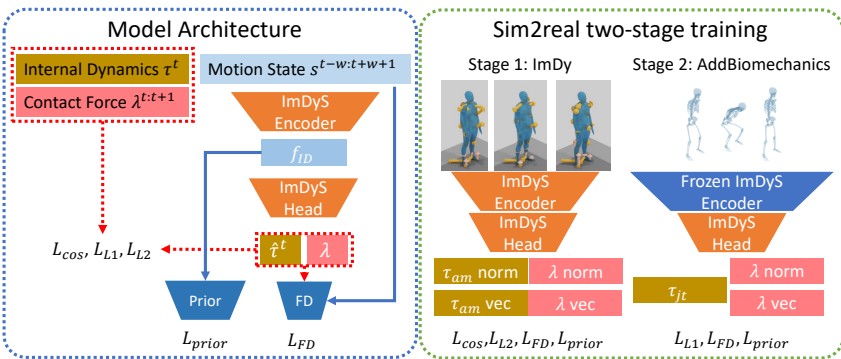

Figure 3: ImDyS overview. Taking a motion transition, ImDyS predicts the internal dynamics and ground reaction forces. Moreover, a prior discriminator is trained with the feature from ImDyS. A two-stage sim2real training curriculum is further designed.

$s_o^t \to s_o^{t+1}$. Thus, the idea could be as simple as using $\pi$ to imitate arbitrary motions in the simulator, then collecting all the **observed** states $s_o$, the applied torques $\tau$, and the full-body GRF $\lambda$.

We adopt AMASS (Mahmood et al., 2019) and KIT (Krebs et al., 2021) as two major data sources. Sequences involving humans interacting with objects other than the ground are excluded, resulting in over 50 hours of motion. All the sequences are re-sampled to 30FPS, with the z-axis as the gravity axis. Then, the sequences are imitated three times by the two primitive policies and the multiplicative policy with a simulation frequency of 60Hz, resulting in over 150 hours of human motion data with dynamics. States including $q, \dot{q}, \beta$ are recorded in synchronous with the torque $\tau$, all restored in the format of SMPL (Loper et al., 2015) if possible. Moreover, GRFs for the whole body are also recorded, resulting in ImDy, a large-scale human motion dynamics dataset.

Detailed statistics of ImDy are demonstrated in Tab. 1. There are three major advantages. First, a considerably larger data scale is **100×** compared to previous efforts with full-body dynamics data, covering a wide span of human motion, which could be hard to acquire in laboratory setups. Second, thanks to the advanced simulator (Makoviychuk et al., 2021), we could include ground reaction forces for the whole body instead of the two feet only like in previous efforts. Finally, we represent humans with SMPL (Loper et al., 2015), increasing availability.

## 4 LEARNING IMDYS

With the collected ImDy, we could address the human inverse dynamics in a full-supervised manner with a data-driven solver ImDyS. In Sec. 4.1, we first introduce the formulation of data-driven inverse dynamics. Then, the proposed data-driven solver is introduced in Sec. 4.2. The overall pipeline of ImDyS is illustrated in Fig. 3.

### 4.1 FORMULATION

Recall the abstraction of ID in Eq. 2, which we rewrite as

$$(\tau^t, \lambda^{t:t+1}) = ImDyS(s^{t-w:t+w+1}). \tag{4}$$

Given the kinematics states from timestamp $t - w$ to $t + w + 1$, ImDyS is required to estimate the internal dynamics $\tau^t$ for the transition from $s^t$ to $s^{t+1}$ and the ground reaction forces $\lambda$ that the subject bears in timestamp $t$ and $t + 1$.

**Motion States** $s$ could be represented by either SMPL parameters, joint angles, joint coordinates, or marker coordinates. However, due to the topology divergence, the conversion among SMPL parameters, joint angles, and joint coordinates is non-trivial with limited performances. To guarantee that ImDyS could be seamlessly adopted to both ImDy and real-world biomechanics data, we adopt marker coordinates as motion state representation for ImDyS. The state $s^t = (m^t, \dot{m}^t)$ is composed of marker coordinates $m^t$ and finite-differentiated velocities $\dot{m}^t$ at timestamp $t$, which are easy to obtain for both ImDy and AddBiomechanics (Werling et al., 2025). Two temporal windows before and after the transition with a length of $w$ are included for contextual information. Notice that human physical properties like height and weight could also be implicitly represented by the markers. The states are canonicalized w.r.t. the heading direction of $s^t$.

**Internal Dynamics** $\tau$. For ImDy, the imposed angular momentum $\tau_{am}$ is adopted for dynamics representation. Notice that in Sec. 3.2, the original sequences are in 30FPS, while the simulation runs at 60FPS. This means for each motion transition $(s^t, s^{t+1})$, two torques were applied sequentially, each for $\frac{1}{60}s$. Predicting both torques is a plausible design choice. However, the second torque is based on the un-recorded mid-state between $s^t, s^{t+1}$. Predicting it involves the forward dynamics from $s^t$ to the mid-state, with increased complexity. To this end, instead of predicting instantaneous torques, we switch to predicting the imposed angular momentum $\tau_{am} \in \mathbb{R}^{(J-1) \times 3}$, the time-accumulation effect of torque, for each motion transition. Thus, the modeling could stay consistent with proper complexity, only needing to sum the two torques up for $s^t, s^{t+1}$ and then multiply it with the delta time. For AddBiomechanics, joint torque $\tau_{jt}$ is adopted for dynamics representation.

**Ground Reaction Forces** $\lambda$. Different from previous efforts (Mourot et al., 2022; Han et al., 2023; Werling et al., 2025) with foot GRFs only, we predict full-body GRF $\lambda \in \mathbb{R}^{J \times 3}$ as in Fig. 1.

## 4.2 DATA-DRIVEN IMDYS

**Model architecture.** With the enormous data scale of ImDy, we would like to keep ImDyS simple. An encoder-head structure is adopted. $s^{t-w:t+w+1} \in \mathbb{R}^{M \times (2w+2) \times 6}$ is first flattened as $\tilde{s}^{t-w:t+w+1} \in \mathbb{R}^{M \times (12w+12)}$ with window size $w$ and $M$ markers. Then, a transformer encoder converts $\tilde{s}$ into ID feature $f_{ID} \in \mathbb{R}^d$, where $d$ is the feature dimension. For prediction, we decompose $\tau_{am}$ and $\lambda$ into magnitudes $|\tau_{am}^t|, |\lambda^{t:t+1}|$ and direction vectors $\vec{\tau}_{am}^t, \vec{\lambda}^{t:t+1}$ and predict each of them with a linear head. $\tau_{jt}^t$ is predicted with another linear head. The final predictions are $\hat{\tau}_{am}^t = |\tau_{am}^t| \vec{\tau}_{am}^t, \hat{\lambda}^{t:t+1} = |\lambda^{t:t+1}| \vec{\lambda}^{t:t+1}$ and $\tau_{jt}^t$.

**Loss terms.** L1 loss, cosine loss, and L2 loss are adopted to optimized the predicted magnitudes $|\tau_{am}^t|, |\lambda^{t:t+1}|$, direction vectors $\vec{\tau}_{am}^t, \vec{\lambda}^{t:t+1}$, and joint torques $\tau_{jt}$ as $L_{mag}, L_{cos}, L_{L2}$ respectively. Besides, a forward dynamics (FD) loss $L_{fd}$ is proposed with an auxiliary FD model to inform the learning with the ID-FD cycle. The FD model takes $s^{t-w:t}, \tau^t = (\tau_{am}^t, \tau_{jt}^t), \lambda^t$ as input, predicts the next-frame joint angles. The FD loss is thus computed with cycle consistency as

$$L_{FD} = |s^{t+1} - FD(s^{t-w:t}, \hat{\tau}^t, \hat{\lambda}^t)|. \tag{5}$$

Finally, we devise a loss term similar to Peng et al. (2021), which encourages the ImDy feature $f_{ID}$ to model physically plausible motion transitions. A linear discriminator takes $f_{ID}$ and outputs a logit indicating whether the motion transition is plausible. To train the discriminator, besides the positive samples from ImDy and AMASS (Mahmood et al., 2019), we propose two negative sample generation strategies. First, $s^{t-w:t+w+1}$ is randomly permuted along the temporal axis. Second, random Gaussian noises are added on $s^{t-w:t+w+1}$. Binary cross-entropy loss is adopted as $L_{cls}$.

**Sim2Real training curriculum** is devised in a simple two-stage manner. In the first stage, ImDyS is trained on ImDy, with the overall loss as $\mathcal{L}_{s1} = \alpha_1 L_{mag} + \alpha_2 L_{cos} + \alpha_3 L_{FD} + \alpha_4 L_{cls}$. In the second stage, we freeze the encoder and train the linear head for joint torques $\tau_{jt}$. The loss is calculated as $\mathcal{L}_{s2} = \alpha_3 L_{FD} + \alpha_4 L_{cls} + \alpha_5 L_{L2}$. Results show that ImDy pre-trained encoder converges fast on AddBiomechanics, indicating that it holds useful knowledge on real-world human dynamics.

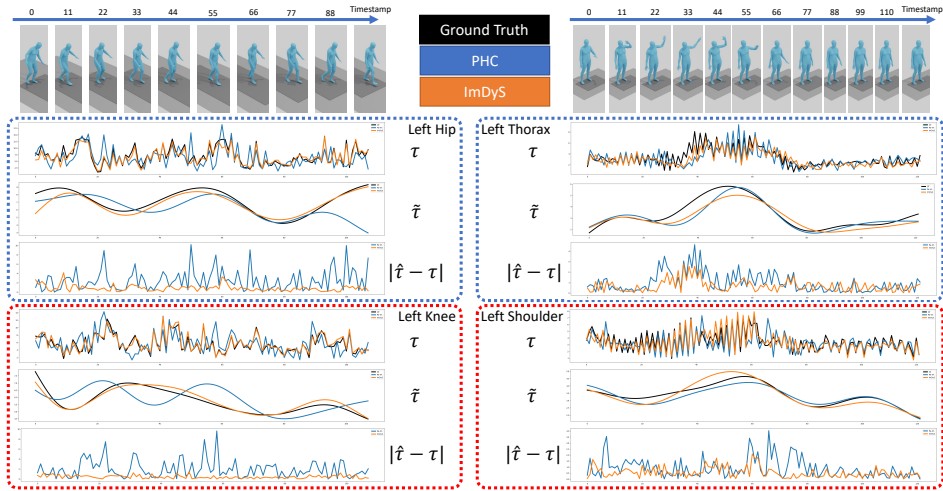

Figure 4: Qualitative results on ImDy. ~ indicates a low-pass filter at 14Hz is applied. A typical gait sample and an arm-waving sample are visualized.

## 5 EXPERIMENTS

### 5.1 IMPLEMENTATION DETAILS

PHC (Luo et al., 2023) adopted the position-control mode implemented by IsaacGym (Makoviychuk et al., 2021), where the imposed torque is calculated differently from the naive PD controller and inaccessible. Therefore, we re-trained the PHC on AMASS (Mahmood et al., 2019) with the effort-control mode, and a naive PD controller was adopted. Training the PHC took approximately 10 days, with a success rate on AMASS of 91.3%. The window size $w$ is set as 2 to keep a short-term motion modeling, which is proven helpful in Sec. 5.3. The encoder of ImDyS is a three-layer transformer with a dimension of 64, ReLU activation, and LayerNorm. The loss weights are set as $\alpha_1 = \alpha_3 = 0.01$, $\alpha_2 = \alpha_4 = \alpha_5 = 1$ to maintain all terms at similar numerical scales for training stability. ImDyS, the prior discriminator, and the FD model are all trained using the AdamW optimizer with a batch size of 2,400 for 140 epochs on ImDy for the first stage. For the second stage, ImDyS is further tuned on AddBiomechanics for only 10 epochs with the same hyper-parameters. When generating negative samples for the prior discriminator, the two strategies are randomly adopted with a positive-negative ratio of 1:1. We split ImDy into a training set of 27,501 sequences and a test set of 3,055 sequences. All the data collection processes and experiments are conducted on a single NVIDIA RTX3090 GPU.

### 5.2 EVALUATION ON IMDY

**Metric.** We calculate the mPJE (mean Per Joint Error) for $\tau$ and $\lambda$ as

$$mPJE_\tau = \frac{1}{J}\sum_{j=1}^{J}|\tau_j - \hat{\tau}_j|_2, \ mPJE_\lambda = \frac{1}{J}\sum_{j=1}^{J}|\lambda_j - \hat{\lambda}_j|_2, \quad (6)$$

where $J$ is the number of joints. The result is further normalized by body weight to align different subjects, with units of $N \cdot m \cdot s/kg$ and $N/kg$. Specifically, the mPJE for the GRF on both feet $mPJE_{\lambda_{lf}}, mPJE_{\lambda_{rf}}$ is also reported.

**Baseline.** Few efforts except IL algorithms are feasible as baselines. To this end, we introduce PHC as a baseline, where the sequences in ImDy are re-imitated by the re-trained PHC. The imposed angular momentums and the GRF obtained via the re-imitation process are adopted as the baseline predictions. With this baseline, we demonstrate the amplification effect from the kinematics error to the dynamics error, thus validating the performance of directly adopting IL for ID.

**Results.** Quantitative results are shown in Tab. 2. PHC produces an mPJPE of 56.13 mm, which is admirable for kinematics but results in high dynamics errors. ImDyS demonstrates considerably better performance. We further visualize two qualitative samples in Fig. 4. Since the raw data could

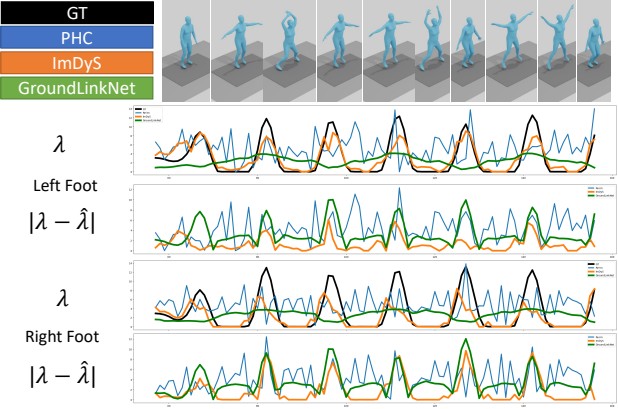

Figure 5: Qualitative results on GroundLink including PHC, GroundLinkNet, and ImDyS. The GRF $\lambda$ for both feet are shown. Surprisingly, ImDyS provides better consistency with the ground truth.

Table 2: Quantitative results on ImDy. mPJE is normalized by the body weight.

| Methods | PHC | ImDyS |
|---|---|---|
| $mPJE_\tau$ $(Nms/kg)\downarrow$ | 0.095 | **0.021** |
| $mPJE_\lambda$ $(N/kg)\downarrow$ | 0.409 | **0.289** |
| $mPJE_{\lambda_{lf}}$ $(N/kg)\downarrow$ | 3.034 | **1.843** |
| $mPJE_{\lambda_{rf}}$ $(N/kg)\downarrow$ | 2.866 | **1.842** |

Table 3: Ground reaction force prediction results on GroundLink.

| Methods | PHC | GroundLinkNet | ImDyS |
|---|---|---|---|
| $mPJE_{\lambda_{lf}}$ $(N/kg)\downarrow$ | 2.362 | 5.423 | **0.986** |
| $mPJE_{\lambda_{rf}}$ $(N/kg)\downarrow$ | 2.636 | 2.891 | **1.149** |

Table 4: Quantitative results on Ad-dBiomechanics.

| Methods | Baseline (150 epochs) | ImDyS (10 epochs) |
|---|---|---|
| $mPJE_\tau$ $(Nm/kg)\downarrow$ | 0.1699 | **0.1626**$_{\downarrow 4.30\%}$ |
| $mPJE_\lambda$ $(Nm/kg)\downarrow$ | 1.0876 | **1.0633**$_{\downarrow 2.18\%}$ |

be jittering, we also filter the predictions with a low-pass filter at 14Hz, denoted as $\tilde{\tau}, \tilde{\lambda}$, which helps reveal the general trend of the predictions. For the gait sample at the left, the imposed angular momentum $\tau$ at the left hip and the left knee are plotted, along with the GRF $\lambda$ at the left toe. We also plot the error between the predicted values and GT values. ImDyS manages to faithfully reconstruct $\tau$ for the left knee and hip with minor errors. Meanwhile, PHC typically produces higher errors due to phase mismatch. As shown, it tends to lag behind the input motion. For GRF, ImDyS also produces reasonable predictions. Besides a typical gait analysis sample, we also demonstrate the performance of ImDyS with an arm-waving motion. The $\tau$ at directly related body segments including the left thorax and shoulder is visualized. ImDyS reproduces the dynamic status with better alignment to GT compared to PHC. Generally, ImDyS produces reasonable ID predictions. A potential issue is the jittering prediction, which is a consequence of the jittering observations in ImDy. However, we show that ImDyS could handle real-world smooth observations well even when trained only on jittering ImDyS. More demonstrations are available in the supplementary video.

## 5.3 EVALUATION ON GROUNDLINK

**Metrics.** GroundLink (Han et al., 2023) provides 1.5-hour motion from 7 subjects with GRF. We adopt subject 7 for evaluation. mPJE$_\lambda$ at both feet normalized by body weight is reported.

**Baselines.** PHC is evaluated similarly to Sec. 5.2. We also report the performance of GroundLinkNet (Han et al., 2023). PHC and ImDyS are not exposed to GroundLink during training, resulting in a **zero-shot** evaluation for ImDyS and the PHC baseline. Also, GroundLinkNet operates on 250FPS motion, while ImDyS and the PHC re-imitation baseline only operate on 30FPS motion. Finally, GroundLinkNet predicts GRF for both feet, while ImDyS and PHC could decouple feet into ankles and toes, and predict GRF separately for each part. We add up the ankle GRF and the toe GRF as the foot GRF. All predictions are re-sampled to 30FPS.

**Results.** Quantitative results are illustrated in Tab. 3. Surprisingly, both ImDyS and PHC manage to outperform the specifically trained GroundLinkNet. We attribute this to the enormous scale of AMASS and ImDy, which is much larger than GroundLink. Moreover, even though ImDyS is trained on simulated ImDy only, it generalizes to real-world data with competitive performance. We visualize the results in Fig. 5, 7. The PHC re-imitation baseline produces jittering predictions similar to Fig. 4. GroundLinkNet, though specifically trained on GroundLink, fails to capture the rapid GRF changes in this jumping jack motion, resulting in a relatively flat output. In contrast, ImDyS surprisingly presents good consistency with GT, and even faithfully reproduces the intense peak GRFs for the left foot for the jumping jack. Besides, the prediction is not as jittering as in Fig. 4, indicating ImDyS could handle real-world smooth data well.

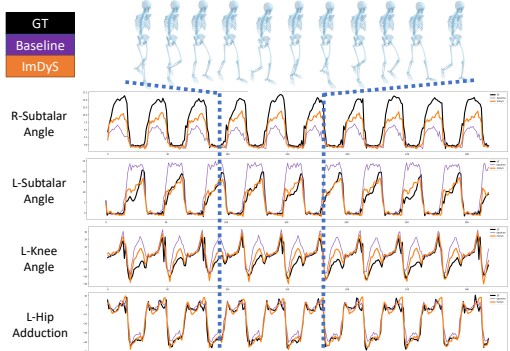

Figure 6: Joint torque predictions on AddBiomechanics.

Table 5: Ablation study on ImDy.

| Methods | $mPJE_\tau$ $(Nms/kg)\downarrow$ | $mPJE_\lambda$ $(N/kg)\downarrow$ | $mPJE_{\lambda_{lf}}$ $(N/kg)\downarrow$ | $mPJE_{\lambda_{rf}}$ $(N/kg)\downarrow$ |
|---|---|---|---|---|
| ImDyS | 0.021 | 0.289 | 1.843 | 1.842 |
| ImDyS-SMPL | 0.011 | 0.272 | 1.746 | 1.764 |
| ImDyS-Joint | 0.014 | 0.273 | 1.755 | 1.775 |
| w/o $L_{FD}$ | 0.023 | 0.302 | 1.962 | 1.976 |
| w/o $L_{cls}$ | 0.022 | 0.294 | 1.884 | 1.891 |

Table 6: Ablation study on AddBiomechanics.

| Methods | ImDyS $w=2$ | ImDyS $w=1$ | ImDyS $w=3$ |
|---|---|---|---|
| $mPJE_\tau\ (Nm/kg)\downarrow$ | **0.1626** | 0.1690 | 0.1720 |
| $mPJE_\lambda\ (Nm/kg)\downarrow$ | **1.0633** | 1.0990 | 1.1030 |

### 5.4 EVALUATION ON ADDBIOMECHANICS

**Metrics.** AddBiomechanics (Werling et al., 2025) is recently proposed with over 50 hours of human dynamics data from 273 subjects. We adopt the armless part of this dataset. We follow the train/test split in Addbiomechanics and report mPJE for the joint torque normalized by body weight.

**Results.** A baseline model trained only on AddBiomechanics for 150 epochs with the same architecture as ImDyS is reported to showcase the generalization from ImDy to real-world dynamics. All data are re-sampled to 30 FPS. Quantitative results are illustrated in Tab. 4. ImDyS outperforms the baseline with faster convergence, indicating the efficacy of Imdys in pre-training and mitigating the sim2real gap. Qualitative results are shown in Fig. 6, 8, where ImDyS shows better alignment with GT and more precise magnitude predictions. More analyses on the relationship between performance, data distribution, and quality are in the appendix.

### 5.5 ABLATION STUDIES

**Different Motion Representations** are evaluated on ImDy in Tab. 5. Though SMPL and joint-based representations perform better, we adopt marker-based representation for its generality.

**Different Loss Terms** are evaluated in Tab. 5. $L_{FD}$ is proven to contribute more than $L_{cls}$.

**Different Window Sizes** $w$ are evaluated on AddBiomechanics in Tab. 6. ImDyS achieves the best balance between rich contexts and conciseness with $w = 2$.

## 6 DISCUSSION

Given the fully simulated nature of ImDy, a reasonable question is the sim2real problem. ImDy could be unnaturally jittering as in Fig. 4. Also, the physical properties of the simulated humanoid differ from those of real humans. Empirically, experiments show that ImDyS generalizes well to real-world data, partially mitigating this gap. The reason could be threefold. First, the jitters are unnatural but still physically plausible given that ImDy faithfully preserves consistent information for the simulated physics phenomena. Second, the small window size of ImDyS prevents it from relying on long-term contexts, where jitters are more salient. Finally, the enormous scale of ImDy is helpful for generalization. To further mitigate the sim2real gap with ImDy is a meaningful goal to pursue. Besides, ImDyS is designed as a first-step baseline to demonstrate the efficacy of ImDy. Introducing more sophisticated designs to regulate the behavior of ImDyS would be preferable. Moreover, ImDy only considers GRF, while other external forces are not involved. Also, interaction with other entities is absent. Exploration of these would be interesting for future works.

## 7 CONCLUSION

Leveraging the inherent resemblance between inverse dynamics and imitation learning, we proposed a novel human dynamics dataset ImDy, which contained over 150 hours of human motion paired

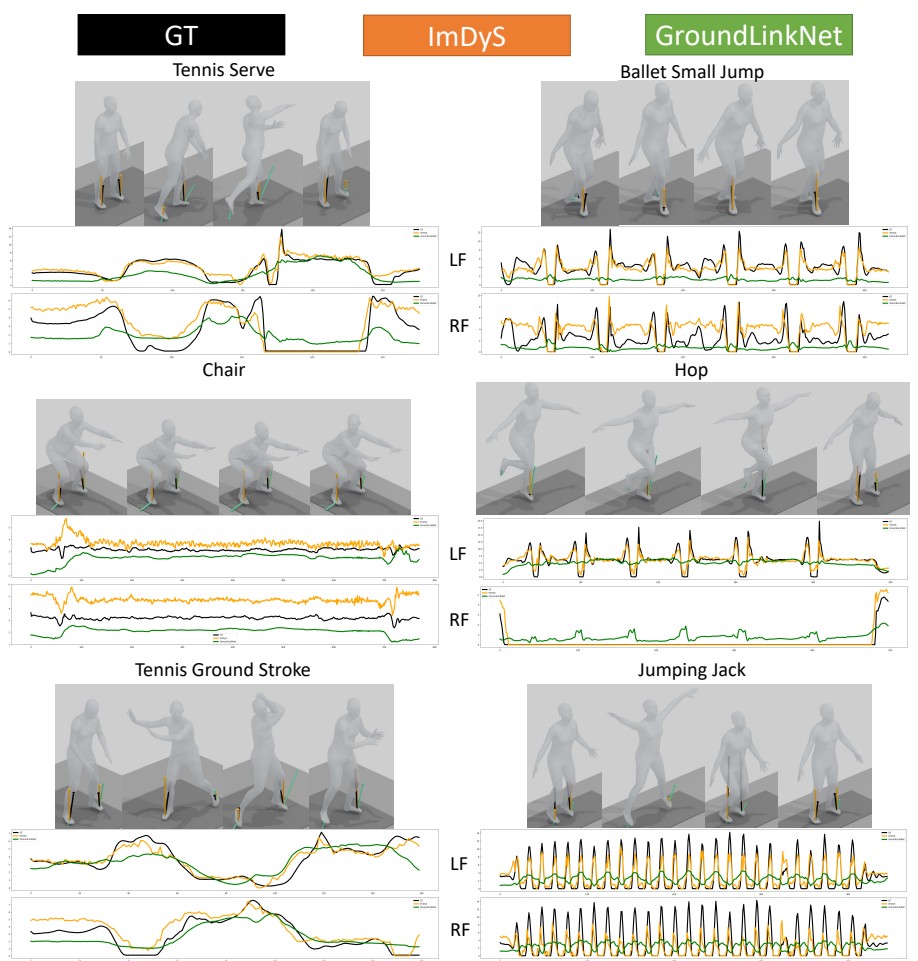

Figure 7: Extensive visualization on GroundLink. ImDyS shows superior alignment with GT for various motions compared to specifically trained GroundLinkNet, showcasing the efficacy of ImDy. Especially, the intense peaks are also reproduced by ImDyS.

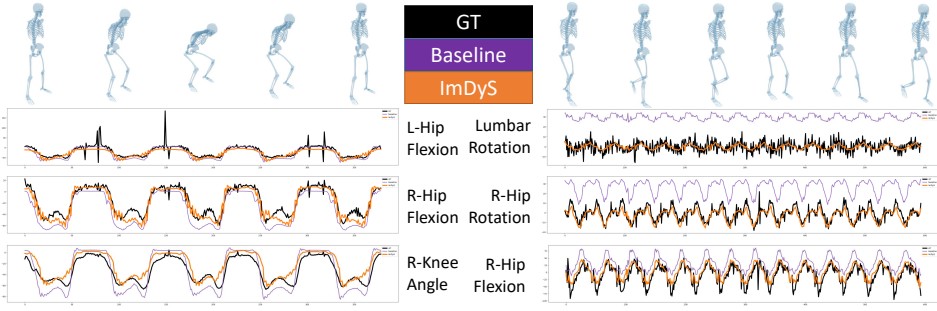

Figure 8: Extensive visualization on AddBiomechanics. ImDyS demonstrates superior performance to the baseline, indicating ImDy's generalization ability.

with full-body driven torques and GRFs from well-developed simulator and imitation algorithms. Based on ImDy, a data-driven human inverse dynamics solver ImDyS is devised to reconstruct the driven angular momentum and contact forces from kinematic observations. ImDyS demonstrated impressive performance on both simulated and real-world data. As a first step toward scalable and easily accessible human inverse dynamics, we hope ImDy can shed new light on the data-driven physical analysis of human motion.

ACKNOWLEDGEMENTS

This work is supported in part by the National Natural Science Foundation of China under Grant No.62306175, CCF-Tencent Rhino-Bird Open Research Fund.

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

APPENDIX

# A    APPLICATIONS OF IMDYS

In this section, we demonstrate some downstream applications of ImDyS.

**Human Work Analysis.** With the predicted $\tau$, we could calculate the work conducted at each joint. Visualizations are in Fig. 9, reasonably revealing the energy flow during human motion.

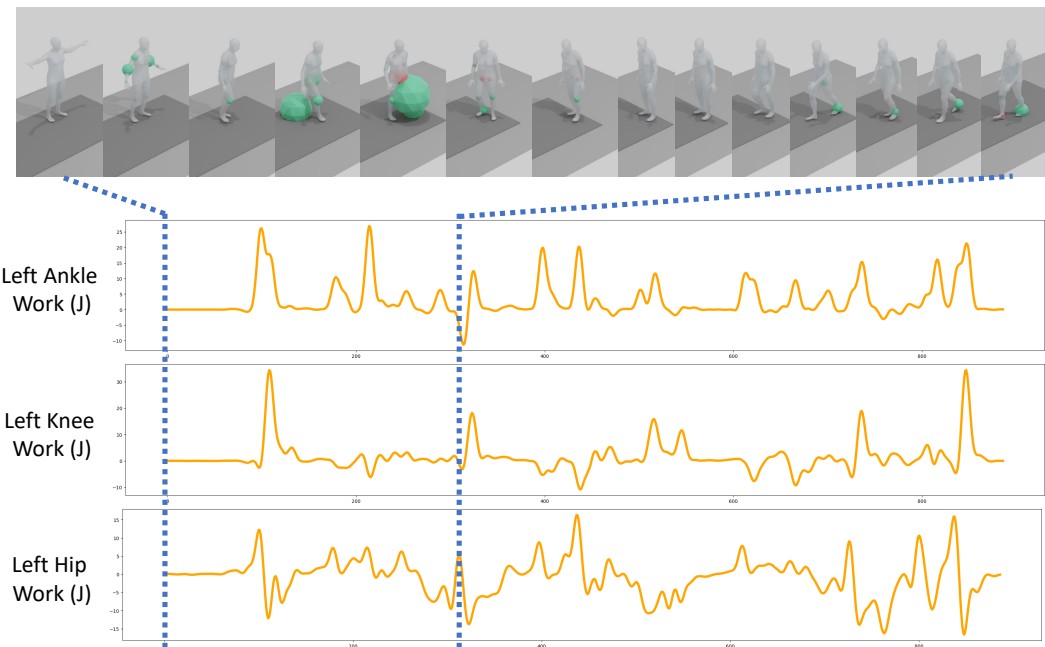

Figure 9: Human work visualization with ImDyS prediction. Green indicates positive work and red indicates negative work.

**Motion Assessment.** Another interesting application of ImDyS is based on the discriminator introduced in Sec. 4.2. Besides facilitating ImDyS learning, it could also assess whether a motion transition is physically plausible as in Fig. 10. Specifically, we adopt ImDyS to assess the motion generated from MDM Tevet et al. (2023). ImDyS reasonably tells when the motion starts to deviate from realism.

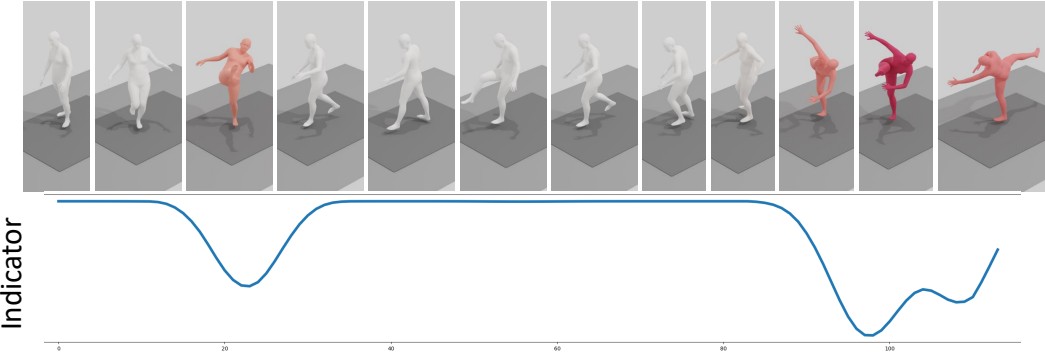

Figure 10: Motion assessment visualization. The motion artifacts are annotated with red with a low indicator value from ImDyS. As shown, ImDyS manages to identify implausible transitions in a kicking motion generated by MDM.

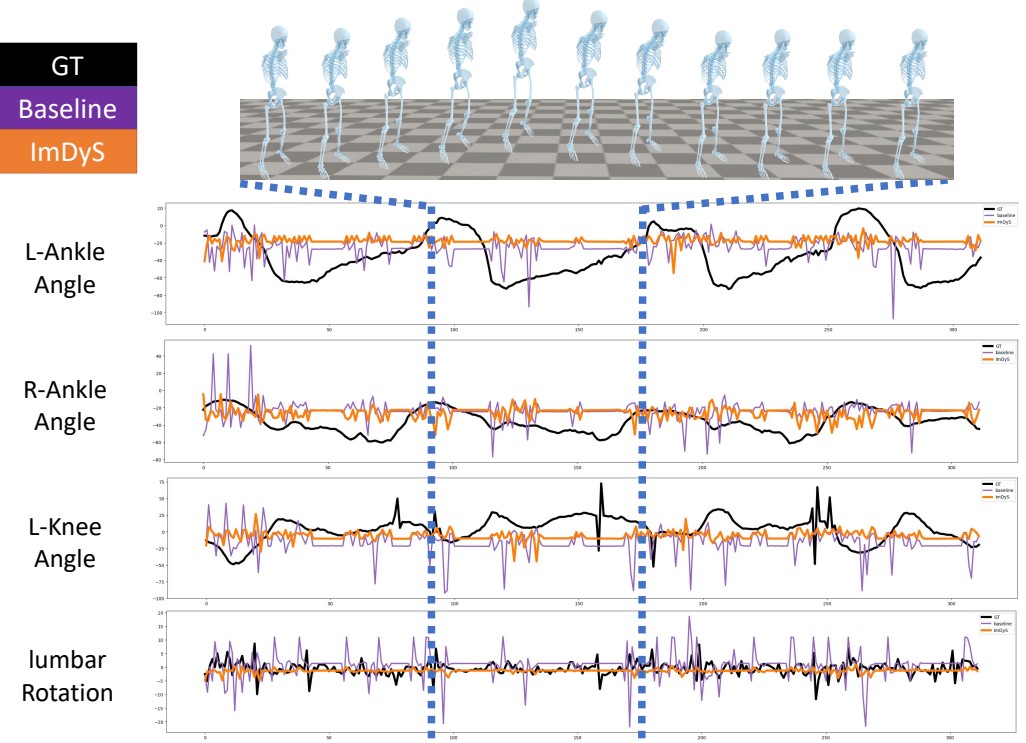

Figure 11: Visualization of a failed joint torque prediction case on AddBiomechanics. For the "jumping" motion, the baseline and ImDyS both perform sub-optimally. Neither of them correctly predicts the joint torques.

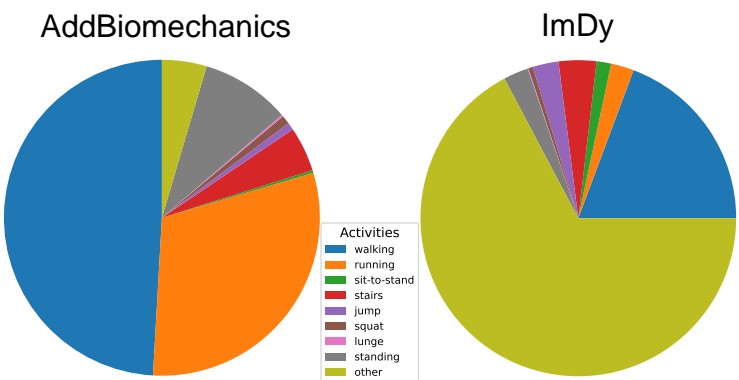

Figure 12: Data distribution of AddBiomechanics and ImDy. Among all activities, walking and running account for over 75% of AddBiomechanics. In comparison, according to the annotations from BABEL (Punnakkal et al., 2021), ImDyS is less imbalanced with better diversity.

## B ADDBIOMECHANICS RESULTS ANALYSIS

We visualize a failure case on the AddBiomechanics dataset in Fig. 11. As shown, neither the baseline nor ImDyS manages to faithfully predict the joint torques for the jumping motion. In the following, we discuss the reasons for the failure.

**Data distribution.** Fig. 12 shows the data distribution of AddBiomechanics (Werling et al., 2025). As shown, over 75% of the data are either walking, running, or standing, which are extremely limited. Though ImDyS is empowered with the diverse ImDy as shown in Fig. 12, it still requires

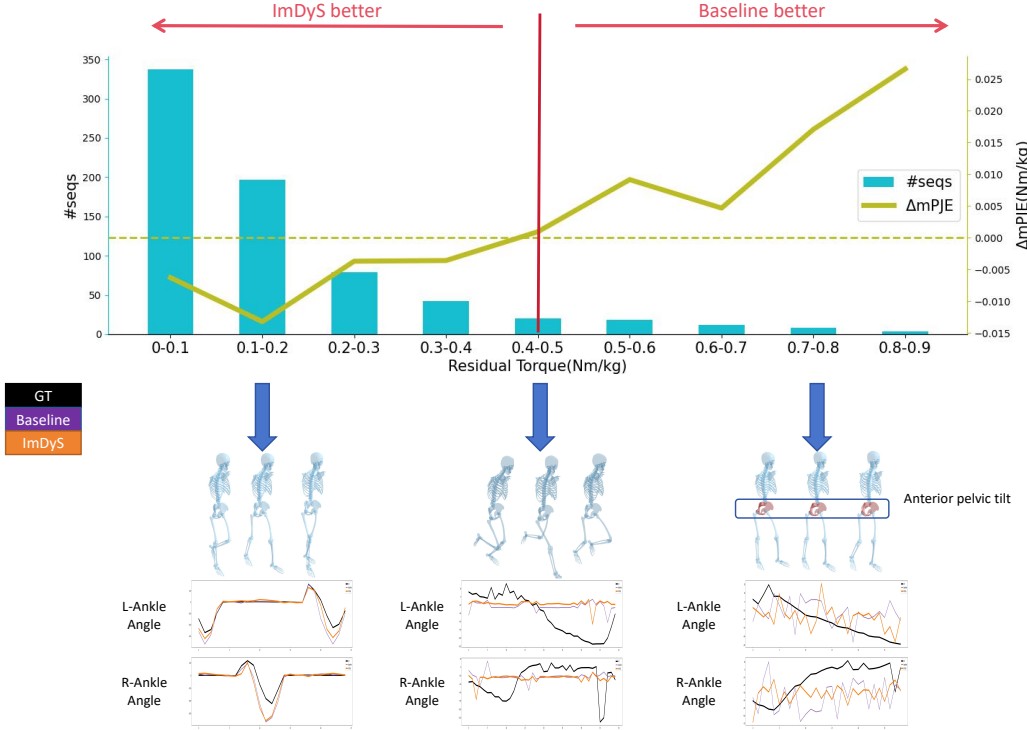

Figure 13: Relationship between data quality and model performance differences. Higher residual torque indicates lower data quality with lower reliability of the optimized GT torques. ΔmPJE is the difference between the mPJE of ImDyS and the baseline. #seqs is the number of sequences. With the residual torque increasing, the baseline provides lower mPJE than ImDyS, indicating the baseline overfits low-quality data. Instead, ImDyS, with the knowledge inherited from ImDy, shows less overfitting for these cases.

data to learn the mapping between simulated torques and real torques for non-gait data. These result in ImDyS' poor performance when processing non-gait data. Further mitigating the limited data issue for non-gait motions would be a meaningful goal to pursue.

**Data quality.** Besides the distribution, the quality is also limited in AddBiomechanics. As shown in Fig. 11, joint torques for some joints (like the lumbar) suffer from unstable optimization with jittering results. According to Werling et al. (2025), 21.2% of AddBiomechnics are classified with clinical-grade high quality (residual torque < 0.1 * body weight * height). There exists a 1.6829 Nm/kg average root residual torque of the optimized GTs in AddBiomechanics, which is considerably higher than the mPJE of ImDyS (0.1626 Nm/kg). We further analyze the relationship between the data quality and the model performances. We adopt residual torques as an indicator of the data quality and calculate $\Delta\text{mPJE}=\text{mPJE}_{ImDyS}-\text{mPJE}_{Baseline}$ of sequences with different residual torques. Notice that higher residual torque indicates lower data quality with lower reliability of the optimized GT torques. Results are shown in Fig. 13. As shown, some samples could suffer from bad kinematics fitting (like the unnatural anterior pelvic tilt in Fig. 13), resulting in less reliable GT optimized joint torques. An interesting phenomenon is that the lower the residual torques are, the better ImDyS performs, which means ImDyS performs better for high-quality samples. This indicates the baseline might overfit low-quality data with high residual torques. Instead, ImDyS, with the knowledge inherited from the large-scale diverse ImDy, manages to resist the negative influences from low-quality samples. We also show how the mPJE of ImDyS changes with data quality in Fig. 14. As shown, the performance of ImDyS degenerates synchronously with data quality.

**Per-Joint Performance Analysis.** It is also noticeable in Fig. 11 that the gap between GT and prediction differs for different joints. To this end, we further analyze the per-joint performance of ImDyS. The per-joint mPJE of ImDyS and the per-joint mPJE of ImDyS in each frame for samples

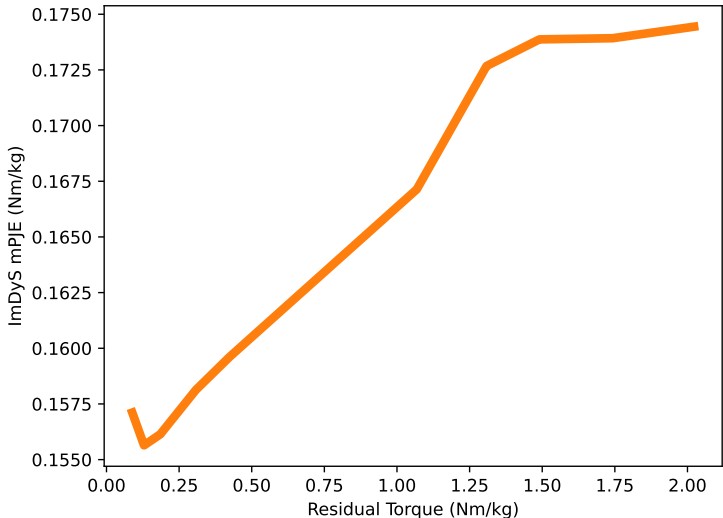

Figure 14: Relationship between data quality and ImDyS performance. Higher residual torque indicates lower data quality with lower reliability of the optimized GT torques. The performance of ImDyS degenerates synchronously with data quality.

Table 7: Per-Joint mPJE of ImDyS for samples with clinical-grade quality.

| Joint Name | Right mPJE$_\tau$ ↓ | | Left mPJE$_\tau$ ↓ | |
|---|---|---|---|---|
| | ImDyS | Baseline | ImDyS | Baseline |
| Hip Flexion | **0.267** | 0.270 | **0.273** | 0.274 |
| Hip Adduction | **0.196** | 0.212 | **0.198** | 0.199 |
| Hip Rotation | **0.087** | 0.109 | **0.083** | 0.098 |
| Knee | 0.193 | **0.188** | **0.195** | 0.209 |
| Ankle | 0.197 | **0.195** | 0.202 | **0.199** |
| Subtalar | **0.069** | 0.071 | **0.079** | 0.084 |
| MTP | **0.0005** | 0.0006 | **0.0007** | 0.0008 |
| Lumbar Extension | **0.328** | 0.339 | - | - |
| Lumbar Bending | **0.255** | 0.255 | - | - |
| Lumbar Rotation | **0.113** | 0.113 | - | - |

with clinical-grade quality (residual torque < 0.1 body weight * height) is demonstrated in Tab. 7. ImDyS manages to improve the performance on most joints compared to the baseline without ImDy, especially for the hips. An interesting phenomenon is that ImDyS performs slightly better on the right half of the body.

## C    SIM2REAL ANALYSIS

We further analyze the Sim2Real effect of ImDy(S) via Fig. 15. An interesting question is the performance of ImDyS without any fine-tuning on AddBiomechanics. Though this could be inapplicable for most joints due to the human model definition discrepancy between Rajagopal's model in AddBiomchanics and SMPL in ImDy, the knee joints in the two models could roughly correspond to each other. Therefore, we visualize the knee torque magnitudes of ImDyS and ImDyS w/o Sim2Real finetuning on AddBiomechanics in Fig. 15. Even without fine-tuning, ImDyS could reproduce the trends of knee torque magnitudes. However, artifacts could also be observed in two aspects. First, ImDyS w/o Sim2Real tends to produce much larger torques. Second, ImDyS w/o Sim2Real could be over-active compared to real humans and ImDyS like in the red circles. The reason could be two-fold. First, the simulation parameters used by ImDy, like mass and inertia, are different from real humans. Second, though the knee joints could roughly correspond, the knee in SMPL has more

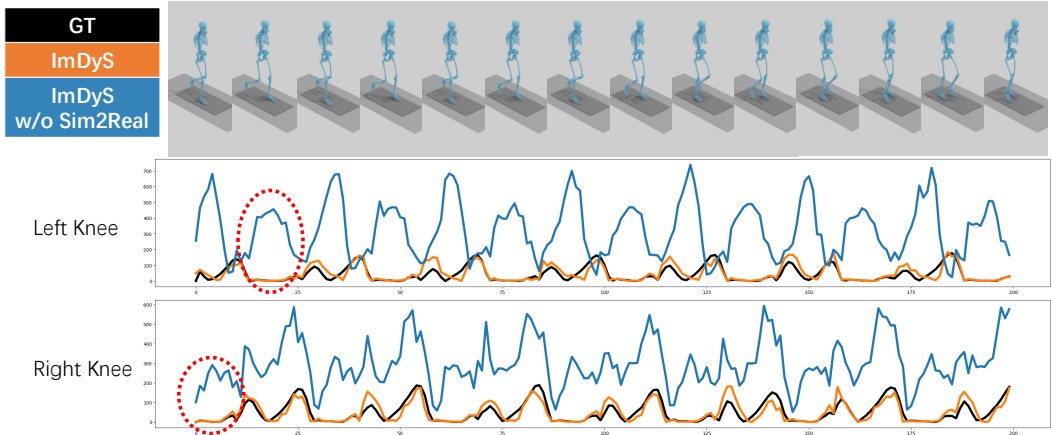

Figure 15: Knee torque magnitude visualization of ImDyS and ImDyS w/o Sim2Real fine-tuning on AddBiomechanics. ImDyS w/o Sim2Real produces larger magnitudes and over-active torques w/o Sim2Real fine-tuning as circled in red.

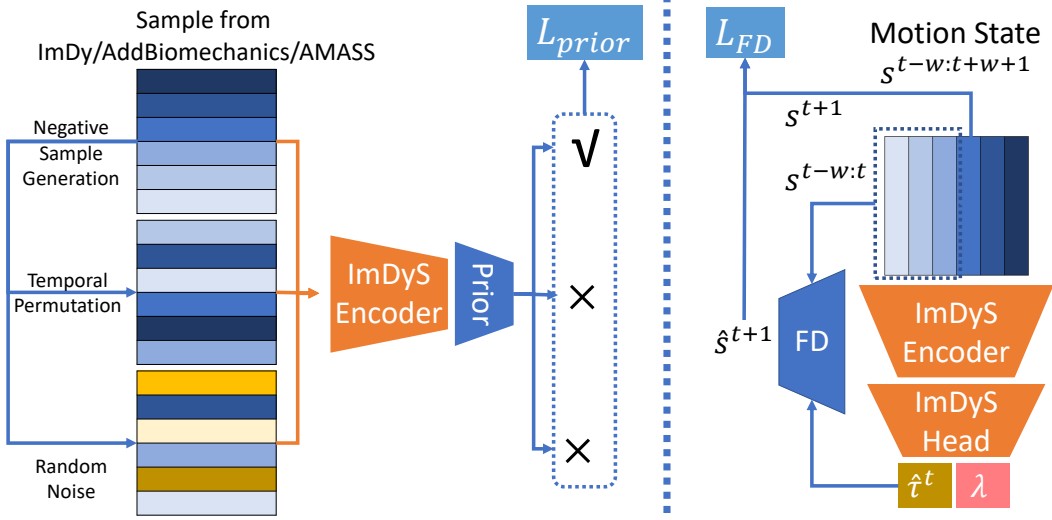

Figure 16: Details of $L_{prior}$ and $L_{FD}$.

DoFs than Rajagopal's model, which might require larger torques to produce similar motions. With the simple Sim2Real fine-tuning of ImDyS, the issues could be alleviated. Further exploration for better Sim2Real performance would be meaningful future work.

## D    DETAILS ON DATA FLOW

Details of the adopted $L_{prior}$ and $L_{FD}$ are illustrated in Fig. 16.

For $L_{prior}$, the input motion state is treated as the positive case, and we generate corresponding negative cases by either temporal permutation or adding random noises. The samples are fed to the encoder, and the prior discriminator predicts whether the sample is positive.

For $L_{FD}$, we first feed ImDyS with motion state $s^{t-w:t+w+1}$, obtaining $\tau, \lambda$. Then, $\tau, \lambda, s^{t-w:t}$ are fed into the FD model, outputing $\hat{s}^{t+1}$. The FD loss is computed as $L_{FD} = |s^{t+1} - \hat{s}^{t+1}|$.

Table 8: Extended results on the FD model on Addbiomechanics.

| Methods | Baseline | ImDyS | Nimble |
|---------|----------|-------|--------|
| RMSE | 0.0302 | 0.0194 | 0.0186 |

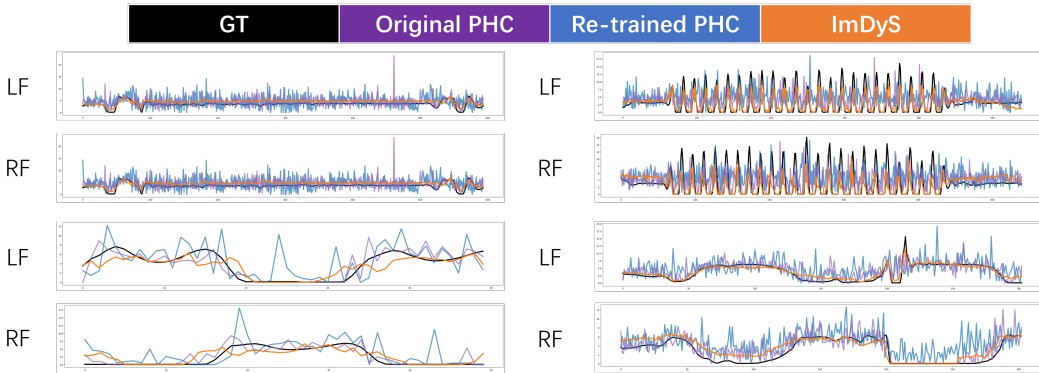

Figure 17: Original PHC on GroundLink.

## E  ANALYSIS ON FD MODEL

We report the marker RMSE of the FD model on the AddBiomechanics test set as Tab. 8. ImDyS noticeably outperforms the baseline trained on AddBiomechanics only, indicating the importance of ImDy pre-training. Moreover, ImDyS is competitive even compared to the differentiable simulator Nimble.

## F  COMPARISON WITH ORIGINAL PHC

Due to the inaccessible torques, we did not include the original PHC as a baseline for ImDy. However, it is noticeable that the original PHC can also conduct GRF prediction. To this end, we also evaluate the original PHC on GroundLink. It provides a left-foot mPJE of 1.559 and a right-foot mPJE of 3.518, which are comparable to re-trained PHC and worse than our proposed ImDyS. Some visualizations are included in Fig. 17. Even without the naive PD controller, the original PHC could suffer from jittering predictions, which could result from the non-perfect contact simulation. In contrast, ImDyS could produce smoother predictions with higher precision.

