# OpenReview forum: "ImDy: Human Inverse Dynamics from Imitated Observations"
_ICLR.cc/2025/Conference — ICLR 2025 Poster_

### Official Review · Reviewer_yxxH · 2024-10-28

**Soundness:** 3
**Presentation:** 3
**Contribution:** 3
**Rating:** 8
**Confidence:** 4

**Summary:**

This paper proposes ImDy, a human inverse dynamics benchmark & dataset that contains 150 hours of motion with joint torque and full-body ground reaction forces obtained from humanoid motion tracking. ImDy provides a scalable approach to obtain more joint-level actuation for human motion as well as ground reaction force grounded in physics simulation. Experiments show that data obtained from ImDy could support learning an inverse dynamics model that has competitive performance on real-world data, showcasing the advantage of having large-scale synthetic dynamics data obtained from simulation.

**Strengths:**

- The proposed ImDy and ImDyS, as far as I know, is a novel attempt at using simulation and humanoid motion tracking to obtain large datasets for inverse dynamics analysis. The results show that the large-scale dynamics data obtained from simulation could be of use in learning models (ImDyS) that perform well on real human performance data.
- A comprehensive analysis of the proposed ImDyS and the baselines is provided. Since this is a relatively new topic, not many prior arts exist. Nonetheless, ImDyS outperforms the baselines and generalizes to real-world data with competitive performance.
- I appreciate the analysis of failure cases, which analyzed the advantages of ImDyS and plausible causes for failure in predicting the joint torques on the jumping motion.

**Weaknesses:**

- Since a dataset is proposed, an analysis of the data composition (like in Figure 12) would be beneficial. Since AMASS also contains many locomotion data, and the PHC with a naive PD controller has a lower success rate, it would be good to see what motions are tracked and included in the dataset.
- ImDys has many details that seem to be omitted. A data flow chart that includes the different losses in the appendix would be helpful. What is the motivation for including the discriminator loss if it does not contribute much to accuracy (as shown in Table 5)?


Minor:

- I would recommend changing the term "imitation learning" to "motion tracking" as "imitation learning" is quite overloaded nowadays.

**Questions:**

Figure 11 visualizes a failed joint prediction case; the plot shows that ground truth torques are quite smooth while the predicted are jittery (before both the ground truth and predicted are jittery). Based on the provided analysis, it is possible the data quality of AddBioMechanics could be the cause here?

---

> ### Author Response · Authors · 2024-11-22
>
> Dear Reviewer yxxH:
>
> First and foremost, we extend our deepest gratitude for your thorough review and helpful advice in improving the paper. We are thankful for recognizing our work as a novel attempt with comprehensive analyses. Herein, we respond to each of your concerns.
>
> **Data analysis.** Thanks for the advice. We update Figure 12 with ImDy data distribution analysis according to BABEL annotations. As shown, ImDy is less dominated by gaits with higher diversity.
>
> **Data flow charts.** Thanks for the advice. We updated a data flow figure in Appendix D, Figure 15.
>
> **Discriminator loss.** Though the accuracy contribution is not much, we show its potential to produce reasonable frame-level motion assessments in Appendix Section A, which is meaningful for motion capture and generation. We believe a more curated design would be even more beneficial.
>
> **Term of "imitation learning".** Thanks for your kind suggestion. By this term, we would like to raise discussions on the extent of "imitation". In detail, IL like PHC primarily focused on imitating "results" (kinematics). We would like to investigate whether they could successfully imitate the "production mechanisms" (dynamics) of human motion. We agree that "imitation learning" is overloaded, and would consider replacing it with "IL-based motion tracking" when applicable.
>
> **Further analysis on Figure 11.** The sample in Figure 11 has approximately 50\% frames with quality lower than clinical grade, which makes data quality a possible cause of failure, especially for the lumbar torques. However, we would identify the data distribution as a major cause of the failure for ankle and knee torques in Figure 11, given the sample is rarely seen in gait-dominated AddBiomechanics. And more obvious cases of quality-related failures are shown in the lower half of Figure 13.

---

> > ### Comment · Reviewer_yxxH · 2024-12-02
> > **Official Response by Reviewer**
> >
> > ` Thanks for the advice. We updated a data flow figure in Appendix D, Figure 15.` You meant Figure 16?
> >
> > Anyway, most of my concerns have been addressed, and I recommend accepting this paper.

---

> > > ### Author Response · Authors · 2024-12-02
> > >
> > > Sorry for the typo! The data flow figure is Figure 16. We are grateful for your recognition of our work. Thanks!

---

### Official Review · Reviewer_xT3W · 2024-11-01

**Soundness:** 3
**Presentation:** 4
**Contribution:** 3
**Rating:** 6
**Confidence:** 3

**Summary:**

In this paper, authors introduce data-driven inverse dynamics solver (ImDyS) that could estimate dynamic states of the person (including joints torque and ground reaction forces) from the observed kinematics. To train ImDyS in a fully-supervised manner, they first propose the data acquisition to collect scalable human dynamics data following the recently proposed imitation learning methods.

They benchmarked their method on both the imitated data (ImDy) and real-world data, AddBiomechanics, which has the measured ground reaction forces and 3D kinematics labels. The results showed that the proposed method is more accurate to estimate human dynamics from both the imitated and real datasets.

Since the authors borrow the concept of imitation learning from the previous work (PHC), I do not see significant novelty in their data acquisition pipeline. However, given their contribution to processing and sharing of the dataset, and the performance of the supervised inverse dynamics module, I think the paper can contribute to the community.

**Strengths:**

The paper's strengths can be summarized as follows:

1) They train an imitation policy that can imitate the large-scale human motion capture dataset (AMASS and KIT) which provides the pairs of human kinematics and dynamics with the larger scale than any other publicly available datasets.

2) They train an inverse dynamics solver on the processed data, resulting in the state-of-the-art performance in estimating ground reaction force and dynamics state of the human (on both imitated and real data).

3) Good visual presentation is appreciated

**Weaknesses:**

Data acquisition part of this paper does not seem to include significant novelty since most of the part were borrowed from the previous work (PHC, Luo et al.,). I would like authors to elaborate what their imitation learning can offer other than PHC. Can we run PHC to collect kinetic/kinematics pairs of AMASS and KIT data? If so, why do we need to use the proposed method for the dynamics data acquisitions?

**Questions:**

1) Connected to the weakness that I pointed above, why does authors' imitation on AMASS have lower success rate than PHC? I would suggest the authors to provide quantitative/qualitative comparisons of two imitation learning schemes (their and PHC), and justify their purpose of the proposed method.

2) It is not clear what can ImDyS bring over the imitation learning. Let's assume we collect kinematics data and want to extract dynamics from it. We can either run ImDyS to estimate dynamics or ImDy policy to imitate the kinematics (where the dynamics will come together). What is the major benefit of using ImDyS in this case? I guess the inference speed but wonder authors' perspective. I would suggest authors to provide comparisons between the proposed IL and ImDyS to highlight the unique values of ImDyS.

---

> ### Author Response · Authors · 2024-11-22
>
> Dear Reviewer xT3W:
>
> First and foremost, we extend our deepest gratitude for your thorough review and helpful advice in improving the paper. We are thankful for identifying our work as well-visualized and meaningful. Herein, we respond to each of your concerns.
>
> **Data acquisition.** We would like to clarify that one of our key contributions is providing initial benchmarks and baselines instead of an imitation algorithm for data acquisition since as pointed out, it is mostly borrowed from PHC. The major difference is covered in Section 5.1 L321-L323, where we retrain PHC with a different control mode. We respectfully emphasize a different view with it. By "imitation", IL like PHC primarily focused on imitating "results" (kinematics) with limited attention on "production mechanisms" (dynamics). The implementations of most IL algorithms validate it, where the IssaacGym function ``set_dof_position_target_tensor()`` without access to torques was adopted for better precision. Instead, we adopt the function ``set_dof_actuation_force_tensor`` with accessible torques. In view of this, ImDy is proposed as a first step toward large-scale and comprehensive human inverse dynamics analysis, with the major originality of interpreting imitation learning algorithms as an ID knowledge base beyond simple control algorithms.
>
> **Imitation success rate.** Related to the first question, the source of it is the different control modes. Also as shown in Figure 4 and our attached video, the data in ImDy suffer from more severe jittering compared to the original PHC. However, even with its jittering quality, ImDy is surprisingly effective for real-world scenarios.
>
> **ImDy compared to IL.**
> - Experimentally, some straightforward comparisons are available in Tab. 2-3 and Fig. 4-5. PHC, as a representative instance of IL, could suffer from phase shift when directly adopted for dynamics estimation in Fig. 4. And for real-world scenarios, PHC could produce jittering predictions in Fig. 5, which might be due to the adopted naive PD controller. The quantitative prediction errors are also considerably outperformed by ImDyS.
> - Theoretically, given the analysis in [1], only a 2-cm marker error could result in a peak ankle plantarflexion moment error of 26.6 N·m, which is inherently inevitable for imitation learning algorithms with the current tracking precision. Also, the sim2real gap hinders IL from being directly adopted for real-world scenarios like AddBiomechanics.
> - Technically, as mentioned, ImDyS could bypass the involvement of simulation pipelines in inference, resulting in higher efficiency and reduced technical complexity.
>
>
> [1] Thomas K Uchida and Ajay Seth. Conclusion or illusion: Quantifying uncertainty in inverse analyses from marker-based motion capture due to errors in marker registration and model scaling. Frontiers in Bioengineering and Biotechnology, 10:874725, 2022.

---

> > ### Comment · Reviewer_xT3W · 2024-11-24
> >
> > Dear authors,
> >
> > I appreciate your thorough responses to my concerns and questions.
> >
> > Based on the authors' explanation, I understand the main advantage of the proposed IL model over the original PHC is that ImDy uses effort-control mode in the imitation pipeline while PHC uses the position-control mode.
> >
> > Where ImDyS is a great baseline for learning-based human ID, I am still hesitant to argue ImDy's contribution as the data acquisition framework as the original PHC is also capable of collecting dynamics data from kinematics (although its quality is worse than ImDy).
> >
> > I would like to ask one more question; when the authors compare PHC with ImDyS, do the authors use the original PHC or a re-trained version of PHC (the one with effort-control mode) as a baseline? Section 5.2 L363:364 is not clear whether the authors use "the original PHC" as a baseline while "a re-trained version of PHC" is used for ImDy data acquisition.

---

> ### Author Response · Authors · 2024-11-25
>
> Dear Reviewer xT3W,
>
> Thanks for your responses. Here we would like to clarify the following points and answer your further questions.
>
> **Data acquisition with original PHC.** We clarify that the original PHC could not be adopted for data acquisition because the torques could not be acquired under position-control mode. The underlying reasons are two-fold. First, as mentioned before, there are no APIs for torque access in position-control mode. Second, the torques applied in position-control mode are computed implicitly in PhysX by a mixed linear complementarity problem (MLCP) solver instead of naive PD controllers [1], between which noticeable gaps exist.
>
> **ImDy as a data acquisition framework.** Here we would like to clarify that the framework of ImDy is composed of an IL-based motion tracking algorithm with torque access (re-trained PHC as an easy-to-use and effective instantiation, with other choices like re-trained PULSE also available) and a high-performance physics simulator (IsaacGym is adopted, but Mujoco could also be applicable). With the submission, we showcase the feasibility of adopting this data acquisition framework for realistic human biomechanics analysis.
>
> **PHC as a baseline.** For all experiments, we use re-trained PHC as the baseline since the original PHC can't be adopted for torques. However, it is noticeable that the original PHC could be used for ground reaction force estimation. Here we include the results as follows. Some visualizations are included in Appendix F and Figure 17, where jittering predictions are shown for the original PHC even with the MLCP solver, which produces smooth kinematics results. In contrast, ImDyS manages to produce smoother predictions with higher precision.
>
> |         | Retrained PHC | Original PHC | ImDyS |
> |---------|---------------|--------------|-------|
> | LF mPJE | 2.362         | 1.559        | 0.986 |
> | RF mPJE | 2.636         | 3.518        | 1.149 |
>
> [1] https://forums.developer.nvidia.com/t/get-torques-in-position-control-mode/209415/21

---

> > ### Comment · Reviewer_xT3W · 2024-11-25
> >
> > Dear authors,
> >
> > Thanks for the further clarification. All my questions are resolved.

---

### Official Review · Reviewer_qgUn · 2024-11-04

**Soundness:** 3
**Presentation:** 3
**Contribution:** 3
**Rating:** 8
**Confidence:** 4

**Summary:**

The paper presents a dataset ImDy (Imitaed Dynamics) with 150 hours of physically simulated humanoid characters imitating kinematic human motions with joint torques and full-body ground reaction forces. The paper claims that ImDy data is unique in its scale (100x more than previous ones), includes more general physics parameters such as the ground reaction forces (GRFs) for the entire body, and is represented in SMPL. Using this ImDy, the paper proposes ImDyS (Imitaed Dynamics Solver), a data-driven humanoid inverse dynamics solver. ImDyS is a simple encoder-head design where a transformer encoder maps the pose state to the ID (inverse dynamics) feature, which is used to predict the dynamics using linear heads. The notable loss designs are the use of the FD (forward dynamics) cycle consistency loss with an FD model trained together to predict the pose state from the dynamics, and a discriminator loss on the ID features, similar to AMP [Peng et al. 2021]. To overcome the sim2real issues, the model is trained in two stages, first with the ImDy data, and then with the data from AddBiomechanics while freezing the encoder and training only the linear head for joint torques.

**Strengths:**

The proposal is a great data and baseline model to advance the data-driven approach to understanding dynamics in human motions. While the baseline model ImDyS is simple, I see enough novel, especially in the loss design with the cycle consistency using the learned FD model and a discriminator loss on the ID features.

**Weaknesses:**

The paper is straightforward and looks good. I appreciate some clarifications in the questions

**Questions:**

Have the authors studied the error in the learned FD model? Is it possible to replace this with a differentiable FD simulation?

I would appreciate it if authors could share visual ablations with various motions, with and without the AddBiomechanics so I can confirm the effect of the sim2real, especially for motions out of distribution from the AddBiomechanics data. It is hard to tell how much jittering still exists just by looking at the mPJE.

---

> ### Author Response · Authors · 2024-11-22
>
> Dear Reviewer qgUn:
>
> First and foremost, we extend our deepest gratitude for your thorough review and helpful advice in improving the paper. We sincerely appreciate for recognizing our work as novel and straightforward. Herein, we respond to each of your concerns.
>
> **FD model analysis.** We report the marker RMSE of the FD model on the AddBiomechanics test set in the following table. ImDyS noticeably outperforms the baseline trained on AddBiomechanics only, indicating the importance of ImDy pre-training. Moreover, ImDyS is competitive even compared to the differentiable simulator Nimble.
>
> | Methods | Baseline | ImDyS  | Nimble |
> |---------|----------|--------|--------|
> | RMSE    | 0.0302   | 0.0194 | 0.0186 |
>
> **Differentiable FD simulation.** Thanks for the insightful comment. Differentiable FD simulation like Nimble is a feasible design choice. However, the adopted simple FD module shows competitive performance compared to Nimble as in the above table. Also, differentiable simulators could introduce extra complexity and computation load to the compact ImDyS. Given these, we did not include Nimble in our pipeline. While we believe the analytical gradients from differentiable simulators are beneficial, we will consider using them in future works.
>
> **More visual samples** are included in the attached video ``rebuttal_video.mp4``. ImDyS predictions on samples from AMASS are visualized, including both walking samples that are similar to AddBiomechanics and some upper-body movements that are not covered in AddBiomechanics. As shown, the jittering is noticeably less significant compared to predictions on ImDy. Also, please refer to the updated Appendix C for further sim2real analysis.

---

> ### Comment · Reviewer_qgUn · 2024-11-25
>
> I thank the authors for sharing the comparison and thoughts on the FD model. Looks all good.
>
> The rebuttal video is just showing the final ImDyS results. I asked for visual comparisons of ImDyS with and without the AddBiomechanics to support the claim that the two-stage training is indeed necessary. Sorry to bring this up a day before the discussion period deadline.

---

> > ### Author Response · Authors · 2024-11-25
> >
> > Thanks for the follow-up clarification! We kindly refer to the updated Appendix C and Figure 15, where the sim2real fine-tuning is further discussed. Due to the human model divergence (SMPL in ImDy and Rajagopal's model in AddBiomechanics), the joint definitions are inconsistent for ImDy and AddBiomechanics. Therefore, the zero-shot quantitative evaluation of ImDyS on AddBiomechanics is invalid. Even though, we visualize a sample for zero-shot ImDyS prediction on AddBiomechanics knee torques, since the knee could be conceptually related. As shown in Fig. 15, zero-shot ImDyS predictions show similar trends with GT. However, the magnitude differs a lot. This is consistent with our recent findings that ImDy tends to be over-powerful compared to real humans. More detailed analyses are updated in Appendix C.

---

> > > ### Comment · Reviewer_qgUn · 2024-11-25
> > >
> > > All I am asking for is the ImDyS results visualized before and after the sim2real fine-tuning. I do not think this has anything to do with the differences in human representation.
> > >
> > > I assume there is some form of retargeting happening to map the AddBiomechanics human representation to SMPL in the process of fine-tuning. Can authors provide more details on this, if not already in the paper or the appendix?
> > >
> > > Appendix C is very informative for understanding the effect of sim2real. Thank you.

---

> ### Author Response · Authors · 2024-11-25
>
> Thanks for the response!
>
> We would like to clarify that Figure 15 exactly visualized ImDyS before and after finetuning. Without sim2real fine-tuning, ImDyS can't be applied to real-world scenarios because it is not informed with realistic joints (Rajagopal's model in AddBiomechanics). We can only conceptually relate joints like the knee as in Figure 15. Also, the sim2real finetuning helps to reduce the over-powerful and over-active phenomenon in ImDyS when applied to real-world scenarios. The human model differences explain the phenomenon, the necessity of sim2real fine-tuning, and how Figure 15 is designed as a visualization of sim2real fine-tuning ablation.
>
> We adopted markers as the motion representation of ImDyS in L267-275, so the retargeting procedure is not involved.

---

> > ### Comment · Reviewer_qgUn · 2024-11-25
> >
> > >Figure 15 exactly visualized ImDyS before and after fine-tuning
> >
> > Yes I get this, but I wanted to see this in videos. Otherwise, I cannot really tell where the artifact is. If ImDyS without fine-tuning shows completely broken motions, that's great. Please show that to clarify the sim2real issue.

---

> > > ### Author Response · Authors · 2024-11-26
> > >
> > > Thanks for the suggestion!
> > >
> > > The supplementary materials include a new video, ``sim2real.mp4``, with 4 animated samples on sim2real fine-tuning ablation. The overpowerful and overactive phenomenon is consistently present across samples without sim2real fine-tuning.

---

> > > > ### Comment · Reviewer_qgUn · 2024-11-26
> > > >
> > > > I thank the authors for their sincere responses. I will upgrade my score to accept.

---

### Official Review · Reviewer_qrg1 · 2024-11-06

**Soundness:** 3
**Presentation:** 3
**Contribution:** 2
**Rating:** 6
**Confidence:** 5

**Summary:**

This paper focuses on human inverse dynamics, aiming to estimate the driving torques of body joints from motion observations. While previous methods have primarily targeted lab settings, this paper employs imitation learning within physics simulators. Building on this approach, the authors created a dataset by importing the AMASS and KIT datasets into Isaac Gym, obtaining contact forces and joint torques based on the previous PHC method. Experimental results also demonstrate improved performance compared to the AddBiomechanics benchmark after fine-tuning.

**Strengths:**

1. Estimating human dynamics from motion observation is an interesting and meaningful direction.

2. The paper is well-written and easy to follow.

3. The proposed strategy is to make human dynamics look effective.

4. Experiments show better performance than previous methods.

**Weaknesses:**

While the proposed method looks promising, I am concerned that its technical contributions may be limited for an ICLR paper. Numerous previous works on physics-aware pose estimation have utilized human dynamics through either non-differentiable physics simulators with reinforcement learning or differentiable dynamics solvers, both of which seem to be more challenging approaches. In these prior works, dynamics estimation often served as an intermediate task, either implicitly or explicitly. In this paper, it seems that only part of the previous methods were adopted, with an increased focus on dynamics estimation. Furthermore, the method employed is also similar to as that of PHC.

**Questions:**

1. KIT is also part of the AMASS dataset, but the paper mentions using the AMAS and KIT datasets. Is there any difference?

2. ImDyS is tuned on AddBiomechanics for 10 epochs. How is the performance without fintuning?

3. I am also curious about whether a better-estimated dynamics is helpful for downstream tasks like human pose estimation.

---

> ### Author Response · Authors · 2024-11-22
>
> Dear Reviewer qrg1:
>
> First and foremost, we extend our deepest gratitude for your thorough review and helpful advice in improving the paper. We sincerely appreciate for recognizing our work as interesting and effective. Herein, we respond to each of your concerns.
>
> **On the technical contributions.** Thanks for your comments. We respectfully clarify that one of our key contributions is providing initial infrastructures, like data acquisition pipelines and baselines, for data-driven human inverse dynamics. As mentioned in the comments and Section 2 L135-146, many previous efforts took dynamics estimation as an intermediate for kinematics-related downstream tasks, with limited analyses and discussions.
> With its wide application in biomechanics, healthcare, robotics, and sports training, we emphasize that dynamics is more than an intermediate step in understanding kinematics. Given this, ImDy is proposed as a first step toward large-scale and comprehensive human inverse dynamics analysis. Its major originality is interpreting imitation learning algorithms as an ID knowledge base beyond simple control algorithms. Therefore, we claimed our major technical contribution as a novel pipeline for human inverse dynamics data collection with original insights and comprehensive analyses of its feasibility for both direct and downstream applications in Section 5 and Appendix A.
>
> **AMASS and KIT.** Thanks! By the mentioned KIT, we mean [1], which is not contained in the original AMASS publication but is included by the AMASS team in later updates. The citation in the paper has been fixed.
>
> **ImDyS w/o AddBiomechanics finetuning.** Due to the human model divergence (SMPL in ImDy and Rajagopal's model in AddBiomechanics), the joint definitions are inconsistent for ImDy and AddBiomechanics. Therefore, the zero-shot quantitative evaluation of ImDyS on AddBiomechanics is invalid. Even though, we visualize a sample for zero-shot ImDyS prediction on AddBiomechanics knee torques, since the knee is the only joint that could be conceptually related. As shown in Fig. 15, zero-shot ImDyS predictions show similar trends with GT. However, the magnitude differs a lot. This is consistent with our recent findings that ImDy tends to be over-powerful compared to real humans. More detailed analyses are updated in Appendix C.
>
> **ImDy(S) for downstream tasks.** As mentioned in Section 2 L135-146, many previous efforts have shown that dynamics estimation is effective for markerless MoCap. We are also working optimistically on the potential of ImDy for downstream tasks as new projects. Moreover, as demonstrated in Appendix Section A, our learned prior managed to detect frame-level motion artifacts, which could be a meaningful tool for motion capture and generation.
>
> [1] Franziska Krebs, Andre Meixner, Isabel Patzer, and Tamim Asfour. The kit bimanual manipulation dataset. In IEEE/RAS International Conference on Humanoid Robots (Humanoids), pp. 499–506, 2021.

---

### Author Response · Authors · 2024-11-22

Dear Area Chairs and Reviewers:

Thanks for your valuable reviews and insightful comments, which have helped us improve our paper. In the initial reviews, most reviewers found our work with a good contribution, presentation, and soundness. We are glad that the proposed human dynamics estimation task is identified as meaningful and interesting. Specifically, we are thankful for identifying our proposal as straightforward (Reviewer qgUn) with great data, a great baseline (Reviewer xT3W), and a novel attempt (Reviewer yxxH). Also, the visualization and analyses are appreciated (Reviewer xT3W, yxxH).

In general response, we would like to clarify that one of our key contributions is to provide initial infrastructures for data-driven inverse dynamics, like data acquisition pipelines and baselines, with the major originality of interpreting imitation learning algorithms as an ID knowledge base beyond simple control algorithms. The motivation is that we believe human dynamics understanding is more than an intermediate for human pose estimation and IL-based motion tracking, with its wide application to robotics [1,2], healthcare [3], and sports training [4].

We have updated our submission as follows, annotated in red:
- Appendix B and Figure 12 are updated with ImDy data distribution visualization.
- Appendix C and Figure 15 are added for further sim2real analysis.
- Appendix D and Figure 16 are added with data flow details.
- Appendix E and Table 8 are added for FD model analysis.
- Appendix F and Figure 17 are added to evaluate the original PHC.
- ``rebuttal_video.mp4`` is added to the supplementary materials with sample visualizations from AMASS.
- Some citations are added and fixed.

Detailed responses to individual concerns are included in the comments. We will keep incorporating the corresponding modifications and expansions to the submission in the revision. Thank you again for your time and efforts paid to your helpful feedback. If you have any further questions, please let us know. We would be delighted to do anything that would be helpful in the time remaining. Thanks!

[1] Figueredo, L. F., Aguiar, R. C., Chen, L., Chakrabarty, S., Dogar, M. R., & Cohn, A. G. (2020). Human comfortability: Integrating ergonomics and muscular-informed metrics for manipulability analysis during human-robot collaboration. IEEE Robotics and Automation Letters, 6(2), 351-358.

[2] Teramae, T., Noda, T., & Morimoto, J. (2017). EMG-based model predictive control for physical human–robot interaction: Application for assist-as-needed control. IEEE Robotics and Automation Letters, 3(1), 210-217.

[3] Yao, S., Zhuang, Y., Li, Z., & Song, R. (2018). Adaptive admittance control for an ankle exoskeleton using an EMG-driven musculoskeletal model. Frontiers in neurorobotics, 12, 16.

[4] Caruntu, D. I., & Moreno, R. (2019). Human knee inverse dynamics model of vertical jump exercise. Journal of Computational and Nonlinear Dynamics, 14(10), 101005.

---

### Meta-Review · Area_Chair_fMzk · 2024-12-14

**Metareview:**

Scientific Claims and Findings:
The paper introduces ImDy, a human inverse dynamics dataset and benchmark containing 150 hours of motion with joint torque and full-body ground reaction force data. The key contributions are:
- A novel data acquisition pipeline that leverages imitation learning algorithms to collect large-scale human dynamics data
- A data-driven inverse dynamics solver (ImDyS) trained on this data that can accurately predict joint torques and ground reaction forces
- Demonstration of the model's effectiveness on both synthetic and real-world data (AddBiomechanics dataset)

Strengths:
+ The paper addresses an important gap in scalable human dynamics data collection and analysis.
+ The approach is well-grounded, using established imitation learning methods and physics simulators.
+ The authors provide thorough analysis including ablation studies, failure cases, and real-world validation.
+ The dataset is significantly larger than existing benchmarks (150 hours) and includes comprehensive dynamics information, which would be of interest to the relevant community

Weaknesses:
- The technical novelty in the data collection pipeline is somewhat limited, as it builds heavily on previous work (PHC).
- Some implementation details of ImDyS were initially unclear, though these were addressed during discussion.
- The sim2real gap remains a challenge

The reviewers has unanimously agreed to recommend the paper to be accepted. I agree with their assessment.

**Additional Comments On Reviewer Discussion:**

Initial Main Comments:

- Reviewer qrg1 raised concerns about technical contribution and similarity to prior physics-based pose estimation works, though acknowledged the method's effectiveness.
- Reviewer qgUn was positive overall, appreciating the data and baseline contributions, but requested clarification on FD model analysis and sim2real performance.
- Reviewer xT3W questioned the novelty in data acquisition pipeline but recognized the dataset's value. Asked for comparisons between ImDyS and imitation learning approaches.
- Reviewer yxxH found it a novel and well-analyzed contribution, but requested more dataset composition analysis and implementation details.

Key changes from author responses include data distribution visualization, sim2real analysis, data flow details, FD model analysis, and evaluation of original PHC. The authors also clarified their main contribution as providing infrastructure for data-driven inverse dynamics rather than novel imitation algorithms.

Given the responses, reviewer qgUn explicitly upgraded their score to "accept" and other reviewers indicated satisfaction with the responses.

---

### Decision · Program_Chairs · 2025-01-22

Accept (Poster)